# The Quality Control of the Automatic Manipulating Process of a Flexible Container When Bulk Materials are Packaged

**Aleksey M. Makarov \*, Oleg V. Mushkin, Maksim A. Lapikov, Mikhail P. Kukhtik and Yuriy P. Serdobintsev**

Department of Automation of Production Processes, Volgograd State Technical University, 400005 Volgograd, Russia; olezhka_93@inbox.ru (O.V.M.); remax777@yandex.ru (M.A.L.); mpkuhtik@gmail.com (M.P.K.); serdobintsevyup@mail.ru (Y.P.S.)

**\*** Correspondence: amm34@mail.ru; Tel.: +7-909-389-89-69

**Abstract:** The quality control of a flexible container (FC) gripped and held by a gripping device during the entire cycle of packaging is an important task in the packaging process of bulk materials in a soft package. Noncontact optical methods of control have been developed and researched for the diagnostics of the automatic manipulating process of a flexible container when bulk materials are packaged in a soft package. Diagnostics of the FC gripping and opening accuracy was carried out herein with the help of machine vision. Processing of the image obtained when the neck of the FC was photographed was carried out by a neural network algorithm, which was made according to a scheme of a perceptron. An automated diagnostics system of the FC gripping and opening accuracy was developed in terms of the obtained algorithm. A control technique based on the algorithm of comparison with a reference was used to reveal the FC gripping and opening defects. This technique consists of preliminary processing of the image obtained from the camera and automatic search for deviations in FC gripping and opening. As a result, a report of defects in the process of FC gripping and opening was obtained.

**Keywords:** automated diagnostics system; image clusterization; flexible container; automatic packaging; machine vision; functional model

## 1. Introduction

Manufacturing quality requirements are constantly on the rise in modern conditions; therefore, it is necessary to realize the modernization of systems and technologies for manufacturing firms in order to satisfy these requirements. It is impossible to check the conformity of one's production without appropriate apparatuses and methods of quality control. Raising the technological level leads to the search for and development of new technological solutions in the field of quality control for manufacturable production.

Quality control of a flexible container (FC) gripped and held by a gripping device during the entire cycle of packaging is an important task when bulk materials are packaged in a soft package. Deviation of the FC neck form from its theoretical model as a consequence of external factors and random errors, for example, discrepancy in the current vacuum level (in a vacuum gripping device (VGD)) from the required one, dimension limits of the FC, and inaccuracy when a gripping device operates, are the defects when control is realized. Considerable theoretical research in the field of quality control of bulk material packaging has been carried out by M. Kreymborg, A.B. Demskij, A.A. Makher Avdi, A.A. Pas'ko, et al.

Noncontact optical methods of control with the use of modern software and hardware suites for image processing using machine vision technologies have been widely adopted for diagnostics of automatic manipulating processes, classification, and other procedures in manufacturing [1,2]. Using machine vision raises the accuracy and quality of diagnostics greatly and allows for correcting the operation of equipment, which increases its efficiency.

The development of new materials on the basis of polymer nanocomposites, which packages are made of, has been carried out along with the development of new methods of quality control for the packaging process of bulk materials [3–5].

Nanotechnologies are a key revolutionary leap of the 21st century in technology and the science of materials. Now, multiphase composites of polymers with nanostructures of organic, inorganic, and polymer additives draw the rapt attention of fundamental scientists and practitioners, first of all because these new materials have improved physico-mechanical, thermal, barrier, optical, and other special properties in comparison with composites of polymers.

The most widespread and studied nano-additives for polymers are, above all, natural ceramics—montmorillonite or vermiculite, which are to be found, for example, in clay minerals and micas, other aluminosilicates, oxides of aluminum, silicon, iron, zinc, magnesium, talc, carbonate, polycarbonate of calcium, coal, aluminum, silver, and also nitrides, carbides, and sulphates of some metals. Layered silicates are inexpensive, accessible, and widely distributed in natural clay minerals and micas. These materials are reduced to nanoscales comparatively easy.

Polymers are generally reinforced with nanoparticles in proportions of 2%–6% by weight, although nanocomposites have been developed with greater percentages of nanoparticles. The properties of the obtained two-phase composites are determined by two main factors:

- the dispersion and distribution of nanoparticles in the polymer matrix;
- interaction between polymer chains and nanoparticles.

The first factor is the key one for securing packaging materials' barrier properties, and the second factor is the key one for increasing the package's physico-mechanical properties.

Different manual and automatic methods can be used for control of the FC gripping accuracy [6–11]. The algorithms which are used in automatic systems of FC gripping control can be divided into using a reference, based on the control of design rules and hybrid algorithms. An algorithm which is based on the use of a reference was chosen herein as the algorithm for the control of the FC gripping accuracy. The theoretical (mathematical) model of the curve form of the FC neck deflection was taken as the reference in this development. Both immediate pixel comparison of the test image with the reference pattern and picking out and following a comparison of informational attributes of elements are possible when the image is compared with the reference. A neural network can implement the comparison process of real images with the reference [12–18].

The choice of architecture in terms of neural networks is one of the most important parts in the development of machine vision systems. There are many factors on which the choice of neural network architecture depends: complexity of the assignment, dynamism of the environment, equipment, etc. A perceptron will fit as a neural network for assignments which do not have a great number of parameters.

The elementary perceptron consists of elements of three types: S-elements, A-elements, and one R-element (Figure 1). S-elements are the layer of sensors or receptors. Each receptor can be in one of two states, resting or active, and it passes a unit impulse to the next layer to associative elements only in the last case.

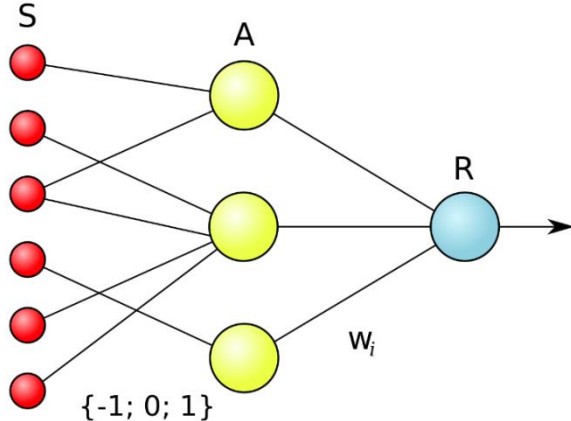

**Figure 1.** The structure of the perceptron neural network.

A-elements are called associative because each such element, as a rule, corresponds with a whole set (association) of S-elements. An A-element is activated as soon as the number of signals from S-elements at its input exceeds some quantity θ. Thus, if the set of corresponding S-elements is situated in the sensory area in the form of the FC neck curve, an A-element is activated when a sufficient number of receptors report the appearance of a "white spot of light" in their neighborhood, i.e., the A-element will be as if associated with the presence/absence of neck curve in a particular area.

Signals from activated A-elements, in turn, are passed to adder R, at which the signal from the ith associated element is combined with coefficient $w_i$. This coefficient is called the weight of the A–R link.

Just like A-elements, the R-element counts the sum of the input signals' values which are multiplied by weights (linear form). The R-element, along with the elementary perceptron, yields 1 if the linear form exceeds bound θ; otherwise, −1 will be the output. The function which is realized by the R-element can be written mathematically in this way:

$$f(x) = \text{sign}\left(\sum_{i=1}^{n} w_i x_i - \theta\right). \tag{1}$$

Methods of cluster or pixel analysis, where an image is broken up into clusters of the same size or number of pixels which are processed in the hidden neurons, can be used for analysis of the image of the FC neck of size $500 \times 500$ pixels. The number of input neurons must correspond with the number of clusters or pixels which are introduced in the neural network; for example, the number of input neurons will be equal to 400 for clusters of size $25 \times 25$. The number of output neurons must correspond with number of aims which the program must reach; there are only two aims in the case of diagnostics of the FC gripping accuracy (a container is ready to use, a container is not ready). The number of hidden neurons, as a rule, corresponds with number of possible variants and their complexity; five variants are enough in this task (the FC has opened correctly, has opened incorrectly, has not opened, has not gripped, has gripped not by all VGDs) and are arranged in one layer.

## 2. The Proposed System of Machine Vision for Diagnostics of FC Gripping and Opening Accuracy

Diagnostics of the FC gripping accuracy was carried out with the help of machine vision. The image which was obtained when the neck of the FC was photographed was processed by the neural network algorithm made according to the scheme of a perceptron. The diagnosis algorithm was represented as a functional model with the help of the Ramus software environment. The Python programming language was used for the development of the automated system software.

### 2.1. The Mathematical Model of the Flexible Container Neck Sagging Curve

It is necessary to understand the mechanism of FC neck sagging when it is opened in order to develop the diagnostic machine vision system for FC gripping and opening. Let us consider the gripping and opening process of a flexible container which lies in a bunker on a level by pneumatic actuating devices, for example, vacuum or jet gripping devices. In this situation, the distance *a* between the external edges of the grippers (Figure 2) must be less than the width B of the container. Then, the FC which is separated from the pile will sag from the unstitched side, opening an internal cavity for its gripping from within.

The contour of the internal cavity of the slightly opened FC neck can be varied by setting the distance *a* between the internal grippers and by changing the number and diameter of gripping devices.

Let us accept the assumption that the FC neck is a closed-loop thread. Then, the equation of the FC neck sagging curve can be calculated by methods of variational calculus as a problem solution to sagging of a thread which is suspended from two sides freely (Figure 2).

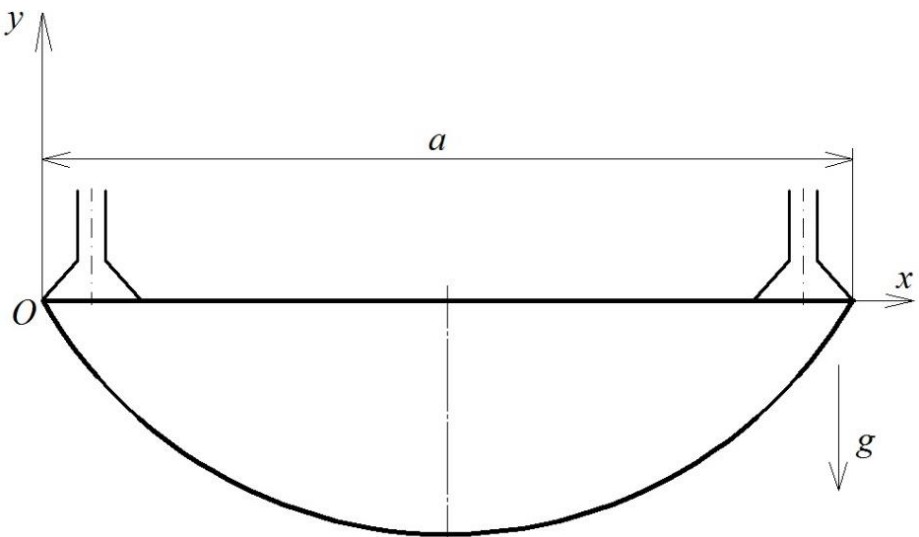

**Figure 2.** Scheme of slightly opened flexible container neck sagging.

Let Lg be the length of the FC neck circumference, so Lg = 2B. Then, (Lg − a) is the length of the sagging neck part. The curve determination problem of the FC sagging neck part is isoparametric. It is required to find the extremum (minimum) of the thread potential energy functional in this problem. We will denote the required function as y = y(x), where x is the abscissa of the function, i.e., the distance of the thread sagging point from the side of the gripper, which is placed at the origin of coordinates O (Figure 2). The potential energy of the thread, which is located in a gravity field, is described by the expression

$$U = \int_0^a \rho \cdot y \sqrt{1 + (dy/dx)^2} dx \qquad (2)$$

when the length of the flexible container neck sagged part equals

$$L_g - a = \int_0^a \sqrt{1 + (dy/dx)^2} dx. \qquad (3)$$

Here, $\rho$ is the specific gravity (N/m$^3$) of the FC cloth length unit.

The extremum of the functional in Equation (2), taking into account Equation (3), is achieved on the condition that

$$Fy - \frac{d}{dx}Fy' = 0, \tag{4}$$

where

$$F = \rho y \sqrt{1 + \left(\frac{dy}{dx}\right)^2} + \lambda \sqrt{1 + \left(\frac{dy}{dx}\right)^2}, \tag{5}$$

where $\lambda$ is an indefinite multiplier.

Equation (4) is the equation of Euler. Solving it for conditions Equations (2) and (3), we get

$$Fy = \sqrt{1 + (y')^2}(\rho y + \lambda), \tag{6}$$

$$Fy' = (\rho y + \lambda)y'\left(1 + (y')^2\right)^{-1/2}, \tag{7}$$

$$Fy - y'Fy' = C_1. \tag{8}$$

Finally, we obtain

$$\frac{(\rho y + \lambda)}{\sqrt{1 + (y')^2}} = C_1. \tag{9}$$

Equation (9) is a differential equation of the FC lower part neck sagging curve. Its solution is given by

$$y = C_1 ch\frac{x - C_2}{C_1} - \lambda, \tag{10}$$

where ch(z) is the hyperbolic cosine of the function; $C_1$ and $C_2$ are arbitrary integration constants, the values of which are determined by the boundary conditions for the concrete quantities Lg and a by solution of the set of equations for the FC suspension points at the length of sagged part (Lg – a).

Let us now find the integration constants and an indefinite multiplier $\lambda$ for the general case when the flexible container is suspended arbitrarily and is held by pneumatic gripping devices (Figure 3). The initial and boundary conditions give the following set of three equations:

$$\begin{cases} C_1 ch\frac{x_0 - C_2}{C_1} - \lambda = y_0; \left(when\, x = x_0 \to y = y_0\right) \\ C_1 ch\frac{x_1 - C_2}{C_1} - \lambda = y_1; \left(when\, x = x_1 \to y = y_1\right) \\ C_1\left(sh\frac{x_1 - C_2}{C_1} - sh\frac{x_0 - C_2}{C_1}\right) = L_g - a. \end{cases} \tag{11}$$

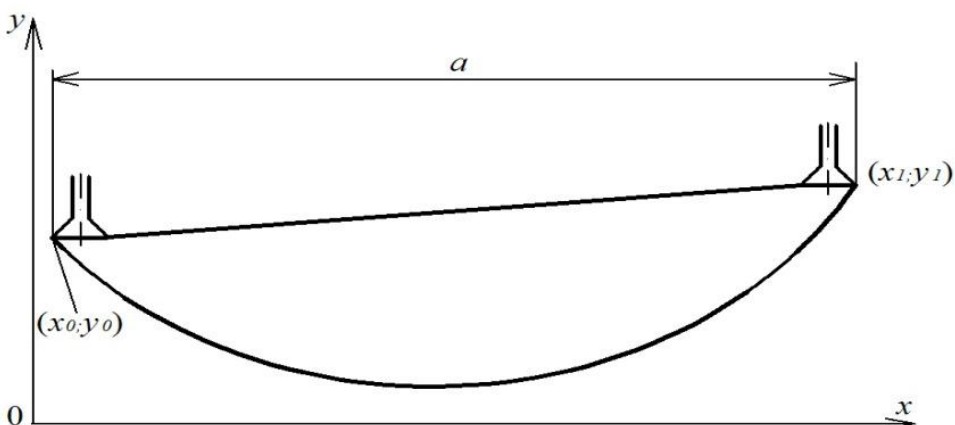

**Figure 3.** General case of holding of the flexible container (FC).

By solving this equation, we obtain

$$2C_1{}^2\left(1 - \text{ch}\left(\frac{x_1 - x_0}{C_1}\right)\right) = \left(y_1 - y_0\right)^2 - \left(L_g - a\right)^2. \tag{12}$$

Evidently, Equation (12) has only one unknown constant $C_1$. We find it by solving this equation by one of the known ways.

We determine $\lambda$ and $C_2$ by substituting $C_1$ into Equation (11).

$$\lambda = C_1 \text{ch}\frac{x_0 - C_2}{C_1} - y_0 = C_1 \text{ch}\frac{x_1 - C_2}{C_1} - y_1. \tag{13}$$

The set of equations for the determination of values $C_1$, $C_2$, and $\lambda$ for the particular case of FC holding, represented in Figure 2, when the planes of the grippers are located on one horizontal line is given by

$$\begin{cases} C_1\text{ch}\frac{-C_2}{C_1} - \lambda = 0; \text{ (when } x = 0 \to y = 0 - \text{left suspension point)} \\ C_1\text{ch}\frac{a-C_2}{C_1} - \lambda = 0; \text{ (when } x = a \to y = 0 - \text{right suspension point)} \\ C_1\left(\text{sh}\frac{a-C_2}{C_1} - \text{sh}\frac{-C_2}{C_1}\right) = L_g - a. \end{cases} \tag{14}$$

We can determine the quantities $C_1$, $C_2$, and $\lambda$ by solving Equation (14) and using graphico-analytical methods. As a result, we obtain

$$C_2 = \frac{a}{2}, \ \lambda = C_1 \text{ch}\frac{a}{2C_1} \cdot C_2 = \frac{a}{2C_1}. \tag{15}$$

$C_1$ is determined from the transcendental equation

$$\text{sh}\frac{a}{2C_1} = \frac{L_g - a}{2C_1}. \tag{16}$$

The graphical solution of the transcendental equation in the program Microsoft Excel for finding $C_1$ is presented in Figure 4. Here, the intersection of the $Z_1$ and $Z_2$ function plots for the left and right parts of Equation (16) is presented, located in the right half of the coordinate plane. The point of their intersection gives the solution of the transcendental equation—value $C_1$. The solution of the equation is in the left half of the coordinate plane, too, but as sh(x) is an odd function, i.e., sh(−x) = −sh(x), both values coincide.

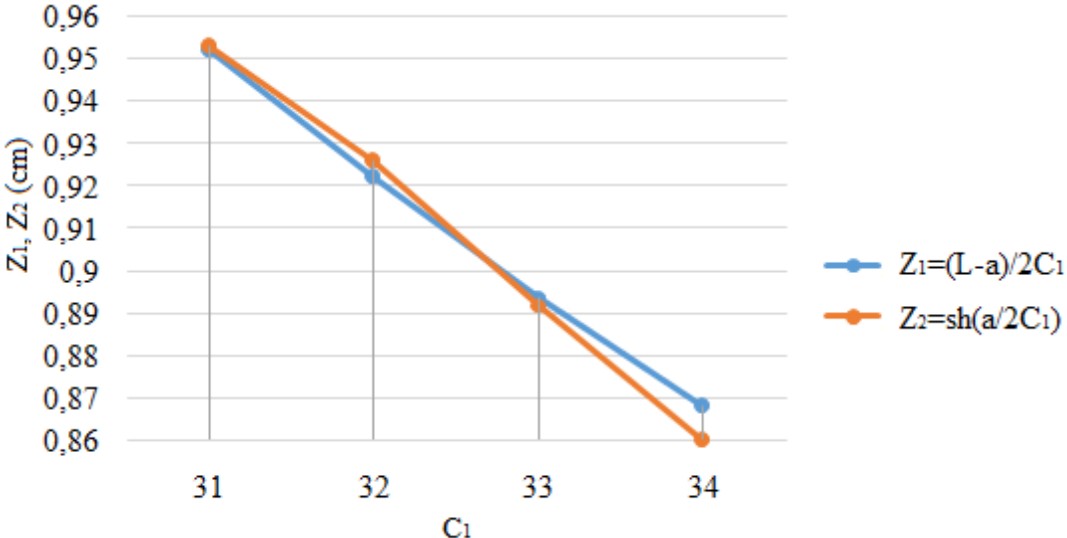

**Figure 4.** Graphical solution of the transcendental equation.

## 2.2. Development of the Algorithm for the Machine Vision System

The diagnosis algorithm of the FC gripping accuracy was developed on the basis of analysis carried out of methods of quality control for FC gripping in the process of bulk material packaging. The present algorithm was represented as a functional model which was made in IDEF0 notation with the help of the Ramus Educational software environment [19–28]. The diagram of first-level decomposition is presented in Figure 5. This diagram consists of four main functional blocks:

- A1, Forming of the image in the area of the FC gripping—getting a picture of the FC gripped by the gripping device;
- A2, Preliminary processing of the image—transformation of the image into black and white, smoothing of shades of grey, partitioning of the image into clusters;
- A3, Diagnostics of the accuracy of the FC gripping—comparison of clusters of the obtained image with clusters of the reference one, recording of comparison results, calculation of probabilities, transmission of calculations on the output neurons of the neural network;
- A4, Forming of the report—forming descriptions of deviations which were revealed in the diagnostics process of the FC gripping accuracy of packaged bulk materials.

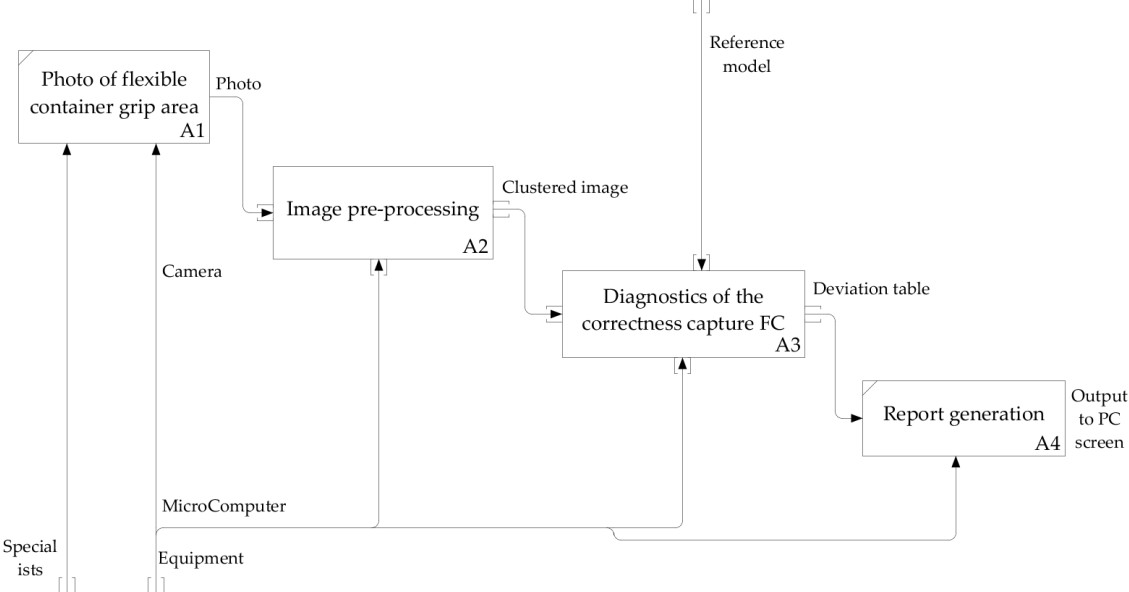

**Figure 5.** The functional model of diagnostics of the FC gripping accuracy (diagram of decomposition).

The diagnostics algorithm of the FC gripping accuracy is represented by a diagram of the second-level decomposition of block A3, shown in Figure 6.

When the image is input to the program of recognition, at first it is transformed into a black and white image to secure the recognition possibility and reduce the number of faults. Further smoothing of shades of grey occurs where superfluous shades are replaced by the nearest requisite ones; this is necessary for avoiding precision loss at the following stages. The image is divided into clusters after accomplishment of the above-listed stages. Afterwards, each cluster is assigned to its input neuron. Then, comparison of the input neuron with the patterns (reference model) occurs. After that, the obtained numerical values of coincidences are recorded in a matrix for the following counting of weights, in particular by the formula of a sigmoid function. Thus, the probability, which is passed to the output neurons, is obtained for each hidden neuron. Since there is only one result of recognition, the greatest result which corresponds to the greatest probability is chosen from the obtained results on the output neurons.

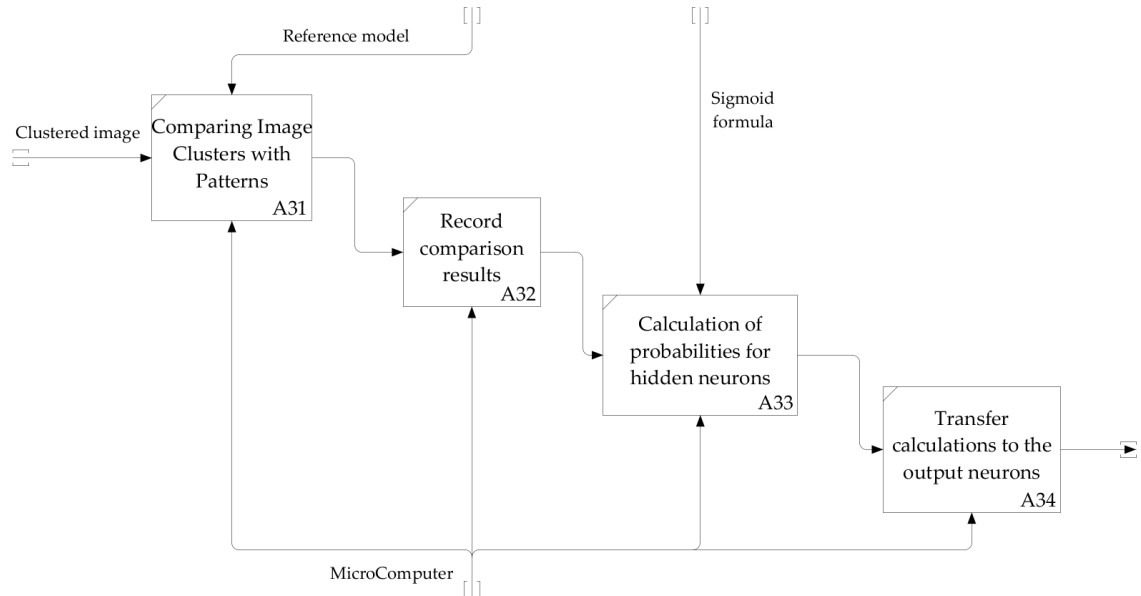

**Figure 6.** The diagram of second-level decomposition of the functional block A3 in diagnostics of the FC gripping accuracy.

### 2.3. Synthesis of the Neural Network Algorithm for Image Processing

A biological neuron is simulated in an artificial neural network (ANN) through an activation function. It simulates the "switching on" of a biological neuron. A sigmoid function is often used as the activation function (Figure 7):

$$f(Z) = \frac{1}{1 + \exp(-z)}. \tag{17}$$

It can be seen from the plot that the function is "activative"—it grows from 0 to 1 as the value of x increases. A sigmoid function is smooth and continuous. This means that the function has a derivative, which, in turn, is a very important factor for the learning of the algorithm.

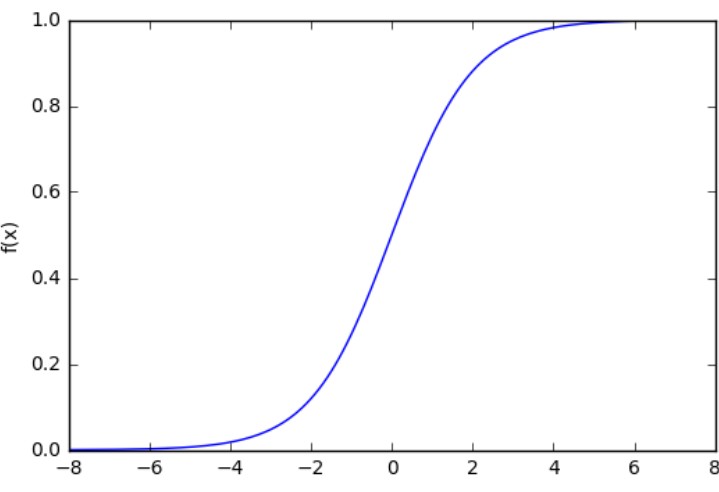

**Figure 7.** The plot of a sigmoid function.

As it was mentioned previously, biological neurons are connected hierarchically in a network, where the output of some neurons is the input for other neurons. We can represent such networks as connected layers with nodes. Each node takes a weighted input, activates the activation function for the sum of the inputs, and generates an output.

A node (Figure 8) is a "location" of the activation function: it takes weighted inputs, adds them, and then enters them in the activation function. The output of the activation function is represented through h.

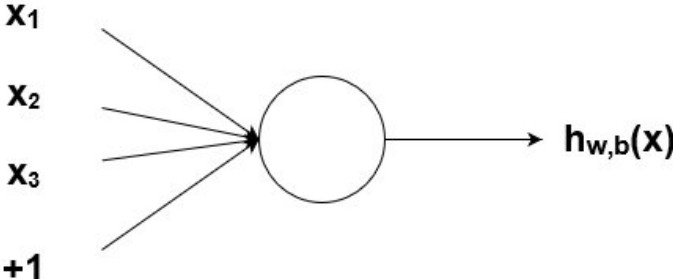

**Figure 8.** A node.

Nonbinary numbers are taken according to weight. Then, these numbers are multiplied at the input and added in the node. In other words, the weighted input in the node is given by

$$x_1 w_1 + x_2 w_2 + x_3 w_3 + b \tag{18}$$

where $w_i$ is a numerical value of weight and b is the weight of the element of displacement at 1.

Weights are extremely important; they are values which will be changed during the process of learning. Switching on weight b makes the node flexible.

Let us consider a simple node which has one input and one output. The input for the activation function in this node is $x_1 w_1$.

One can see in Figure 9 that when the weight changes, the slope level of the activation function plot changes as well. This is comfortable if different densities of correlations between inputs and outputs are modeled. It is necessary to use displacement in order to change the output only on x (Figure 10). It is possible to vary the time of node activation by changing the weight of displacement b (Figure 11). Displacement is very important in cases when it is necessary to simulate conditional relations.

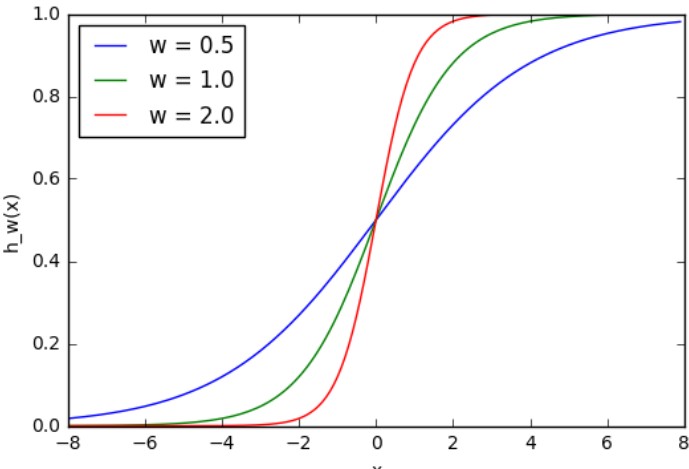

**Figure 9.** A sigmoid function which describes the variation of slope depending on the displacement weight.

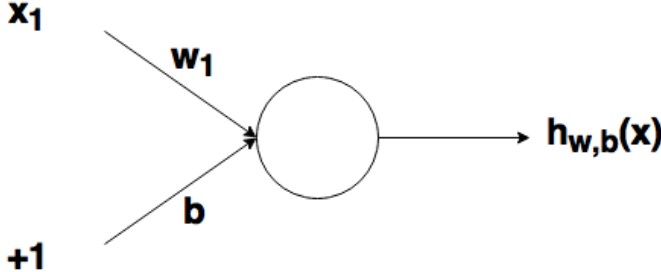

**Figure 10.** A node with displacement at the input.

The creation of a neural network for diagnostics of an input image is based on the perceptron structure (Figure 1). In total, 400 nodes are necessary for the input layer to cover 400 pixels of an image, an output layer with two nodes, and the hidden layer in a network.

Let us declare a simple list in the Python language which determines the structure of the network.

nn_structure = [400, 30, 2]

The sigmoid activation function is used; correspondingly, it is necessary to declare this function and derivative as well.

```
def f(x):
    return 1 / (1 + np.exp(−x))
def f_deriv(x):
    return f(x) * (1 − f(x))
```

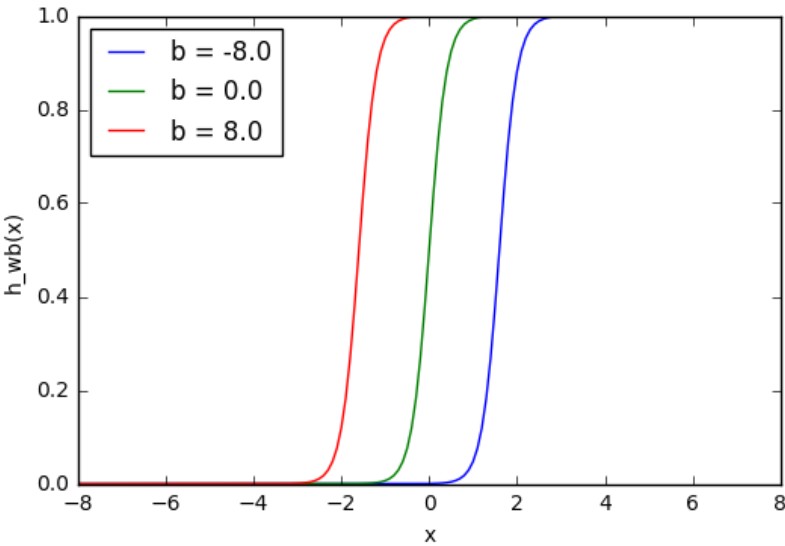

**Figure 11.** A sigmoid function with displacement.

Let us initialize weights $w^{(l)}$ for each layer. Then, $\Delta w$ and $\Delta b$ are assigned, the initial values of which are zero. The step of gradient descent is $\alpha$ for instances from 1 to m. A feedforward process is activated through all nl layers. The output of the activation function is saved in $h^{(l)}$. Value $\delta^{(nl)}$ of the output layer is searched. $\Delta W^{(l)}$ and $\Delta b^{(l)}$ are refreshed for all layers. The process of gradient descent is then activated:

$$w^{(l)} = w^{(l)} - \alpha \left[ \frac{1}{m} \Delta w^{(l)} \right] \tag{19}$$

$$b^{(l)} = b^{(l)} - \alpha \left[ \frac{1}{m} \Delta b^{(l)} \right]. \tag{20}$$

The first stage is the initialization of weights for each layer. Let us use the Python dictionaries for this. Random values are given to weights to ensure the accuracy of neural network operation at the

time of learning. The function random_sample from the library numpy was used for the assignment of random values. The code looks as follows:

```
import numpy.random as r
def setup_and_init_weights(nn_structure):
    W = {}
    b = {}
    for l in range(1, len(nn_structure)):
        W[l] = r.random_sample((nn_structure[l], nn_structure[l-1]))
        b[l] = r.random_sample((nn_structure[l],))
    return W, b
```

Further, zero initial values were assigned to two variables $\Delta w$ and $\Delta b$.

```
def init_tri_values(nn_structure):
tri_W = {}
tri_b = {}
    for l in range(1, len(nn_structure)):
tri_W[l] = np.zeros((nn_structure[l], nn_structure[l-1]))
tri_b[l] = np.zeros((nn_structure[l],))
    return tri_W, tri_b
```

Further, a feedforward process was activated:

```
def feed_forward(x, W, b):
h = {1: x}
    z = {}
    for l in range(1, len(W) + 1):
if l == 1:
node_in = x
        else:
node_in = h[l]
        z[l+1] = W[l].dot(node_in) + b[l] # z^(l+1) = W^(l)*h^(l) + b^(l)
        h[l+1] = f(z[l+1]) # h^(l) = f(z^(l))
return h, z
```

Output layer $\delta^{(nl)}$ and value $\delta^{(l)}$ in hidden layers were found for the activation of back propagation:

```
def calculate_out_layer_delta(y, h_out, z_out):
    # delta^(nl) = -(y_i - h_i^(nl)) * f'(z_i^(nl))
    return -(y-h_out) * f_deriv(z_out)
def calculate_hidden_delta(delta_plus_1, w_l, z_l):
    # delta^(l) = (transpose(W^(l)) * delta^(l+1)) * f'(z^(l))
    return np.dot(np.transpose(w_l), delta_plus_1) * f_deriv(z_l)
```

All stages are united into one function:

```
def train_nn(nn_structure, X, y, iter_num=3000, alpha=0.25):
    W, b = setup_and_init_weights(nn_structure)
cnt = 0
    m = len(y)
avg_cost_func = [[]]
print('Beginning of gradient descent for {} iterations'.format(iter_num))
while cnt 1:
                delta[l] = calculate_hidden_delta(delta[l+1], W[l], z[l])
            # triW^(l) = triW^(l) + delta^(l+1) * transpose(h^(l))
tri_W[l]+=np.dot(delta[l+1][:,np.newaxis], np.transpose(h[l][:,np.newaxis]))
                # trib^(l) = trib^(l) + delta^(l+1)
tri_b[l] += delta[l+1]
```

```
# activates gradient descent for weights in each layer
for l in range(len(nn_structure) - 1, 0, -1):
        W[l] += -alpha * (1.0/m * tri_W[l])
        b[l] += -alpha * (1.0/m * tri_b[l])
# completes calculations of overall evaluation
avg_cost = 1.0/m * avg_cost
avg_cost_func.append(avg_cost)
cnt += 1
    return W, b, avg_cost_func
```

As the limit of the gradient descent operation is unknown, let us activate the function with a fixed number of iterations and observe how the overall function of evaluation with progress in learning is changed. In each iteration of gradient descent, each training instance (range (len (y)) is searched and the feedforward process is activated, then further to back propagation. The stage of back propagation is iterated through the layers, starting from the output layer to the beginning—range (len (nn_structure), 0, 1). The average evaluation is found on the native layer (len (nn_structure)). Values Δw and Δb with mark tri_W and tri_b are refreshed for each layer except native.

After operations on all training instances are completed and values tri_W and tri_b are accumulated, gradient descent is activated and the values of weights and displacements are changed.

After completion of the process, the obtained weight and displacement are returned with the average evaluation for each iteration. Further, the function is called:

W, b, avg_cost_func = train_nn(nn_structure, X_train, y_v_train).

It can be observed that the overall evaluation function decreased after iterative operation of gradient descent (Figure 12).

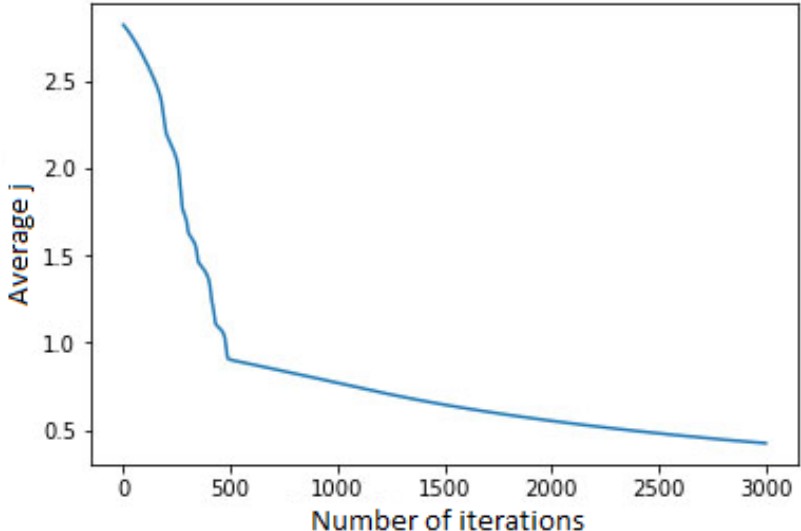

**Figure 12.** The plot of dependence of the overall evaluation function on the number of iterations.

*2.4. Development of the Automated System for FC Gripping and Opening Control*

A system of automatic diagnostics of FC gripping accuracy was developed on the basis of the algorithm (Figures 5 and 6). This system controls the correspondence of the FC neck form to its theoretical model and the presence or absence of the FC at the gripping device. A block diagram of the automated diagnostics system for FC gripping accuracy is presented in Figure 13. This diagram consists of three levels:

- sensor level, which is represented by a Raspberry Pi Camera Board v2.1; photographing of the FC gripping area was carried out using this camera;

- controller level, the architecture of which includes a single-board Raspberry Pi 3 computer and 5V power supply;
- supervisory level, which is represented by the automated workstation (AWS) of the operator, linked with the microcontroller via Ethernet protocol.

Raspberry Pi are considered the most multifunctional mini-PCs. Their advantage lies in the great number of different expansions which have been made for them specially. It is necessary to obtain an image through a minimal number of intermediaries for the possibility of recognition. The Raspberry Pi Camera Board v2.1 was chosen for its effective operation with the Raspberry Pi 3.

The camera is equipped with an 8 megapixel Sony IMX219 Exmor sensor which allows for capturing, recording, and relaying video in 1080p, 720p, and VGA formats. The maximum resolution of frames is $3280 \times 2464$ pixels for photographs.

The more sensitive a camera is, the clearer the images obtained, even in low-light conditions. The quality of an image depends on the camera sensitivity and image resolution. The larger the surface of a sensor itself and separate pixels, the more light gets to it during exposure. Light is transformed into signals which are used by the sensor for the creation and processing of image data. The larger this square is, the higher the signal/noise ratio is; this concerns pixels of large size—3.5 μm and larger—in particular. A value of 42 dB is regarded as a good result.

A large sensor includes a large number of pixels and hence secures higher resolution. The real advantage consists of the fact that separate pixels will be large enough to secure a high signal/noise ratio in contrast to sensors of smaller size and, consequently, of smaller square, for which pixels of smaller size are necessary.

Nevertheless, even big sensors with large numbers of pixels of big size can secure high-quality images only on the condition that a suitable optical system is used. Their potential will be realized fully only with the choice of an appropriate lens that is capable of reproducing such high resolution.

The chosen camera with an 8 megapixel sensor suits our objective. Decreasing the camera sensor pixel number would lead to a decrease in diagnosis precision, and an increase would lead to small growth in the precision of the obtained pictures, but it would not lead to an appreciable rise in diagnosing precision.

Using a multicamera system at large-scale manufacturing facilitates the control of production quality, and it is required when the control of several parameters' quality is necessary. However, using several cameras will not lead to an increase in diagnosing precision in the considered application of a machine vision system.

Special video input CSI (camera serial interface) was used for the connection of the module as it reduces CPU load considerably in comparison with connection of cameras via USB. The automated workstation of the operator was represented by an MSI Gl62m 7rex laptop computer.

Since the developed machine vision system is proposed to be built in, the interface, which secures interaction between the camera-based module and the host system, plays an important role in the architecture of such a system. For example, a USB 3.0 interface with Plug-and-Play technology can be used. LVDS (low-voltage differential signaling) interfaces are worthy of notice as well. Nevertheless, the CSI interface will be an optimal choice in most cases.

From a technological point of view, CSI allows for the development of extremely compact systems in which flat flexible cables can be used for connection between boards. Image data can be transmitted directly from a camera-based or sensor-based module to the processor when a CSI interface is used. Unlike in the case of USB 3.0, it eliminates the need for intermediate processing or data conversion: data are transmitted "directly", which has an effect on processor resource consumption. Corresponding hardware components (for example, a microcontroller) are also not required. It becomes possible to develop built-in systems with extremely low resource consumption.

Integration of the present algorithm into existing control systems of the packaging process (for example, in control programs on programmable logic controllers (PLCs)) will allow accurate diagnosing of the preliminary opening of a flexible container following feed under a loading spout

and filling with bulk material in a real-time mode; this will allow considerably increased reliability of the process due to realizing repeated FC gripping in the case of incorrect or insufficient opening, which will lead to increased efficiency and reduced rejection rate when packaging equipment is operated.

The developed system supports the most popular open standards data transfer protocol, o- industrial Ethernet, which secures the possibility of reliable connection to PLCs and a wide spectrum of automation devices along with simple adjustment of the interface by point-and-click; this facilitates the configuration and service of all user machine vision systems from any computer in a network and archivation of data and images on a file server as well.

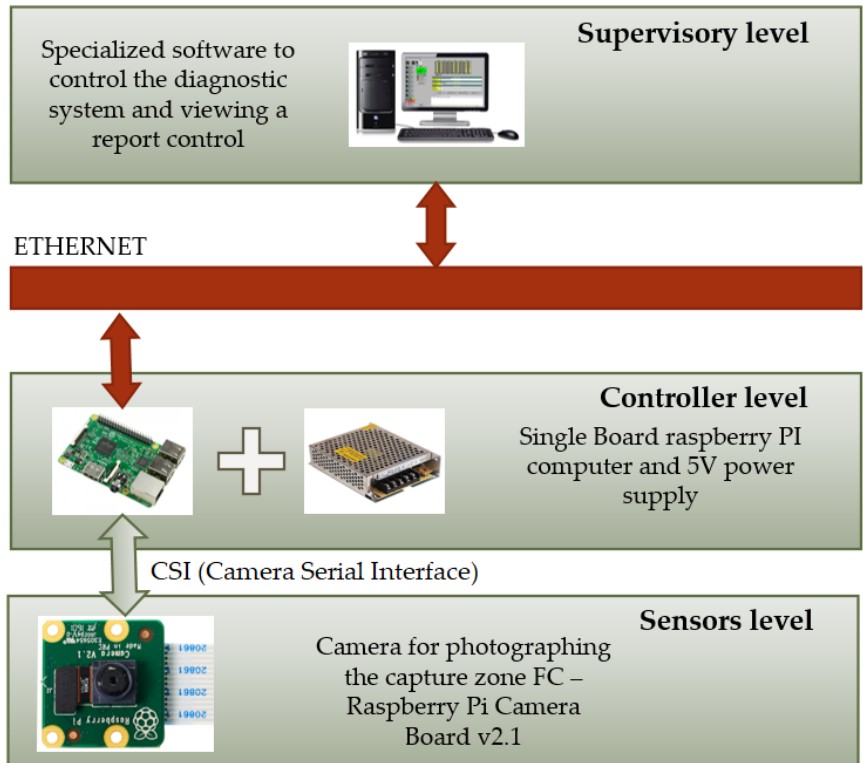

**Figure 13.** A block diagram of the automated diagnostics system for FC gripping accuracy.

The development of machine vision systems for the control and diagnostics of processes of the packaging and packing of foodstuffs, including bulk materials, was presented in [8,11]. However, they did not consider the possibility of scaling, i.e., readjustment on various pieces of technological equipment. Besides this, available systems are confined to the environmental conditions of the technological process, which influences the precision of their performance. All these disadvantages were considered during the development of the represented machine vision system, and the obtained system was constructed using a neural network algorithm to secure the system's self-learning capability.

## 3. Materials and Methods

Experimental research was carried out to check the accuracy of the developed algorithm's operation (Figure 14). It was conducted using a unit for the automatic packaging of bulk materials in an FC (Figures 15–17). The unit includes the gripping mechanism (1), which comprises a bilateral pneumatic cylinder (2) and a vacuum gripper (3) consisting of central bars (4) and end bars (5) (sections). The vacuum gripper (3) is attached to a U-shaped lever (6), which is connected to the rod of the pneumatic cylinder (2). The pneumatic cylinder (2) is jointed to the frame (7) and can rotate around the axis which passes through the pneumatic cylinder center parallel to its foundation. In order to provide the rotation, a pivot (8) is rigidly attached to the housing of the pneumatic cylinder (2). The pivot (8) is jointed to the rod of the bilateral pneumatic cylinder (9), which is mounted on the frame (7).

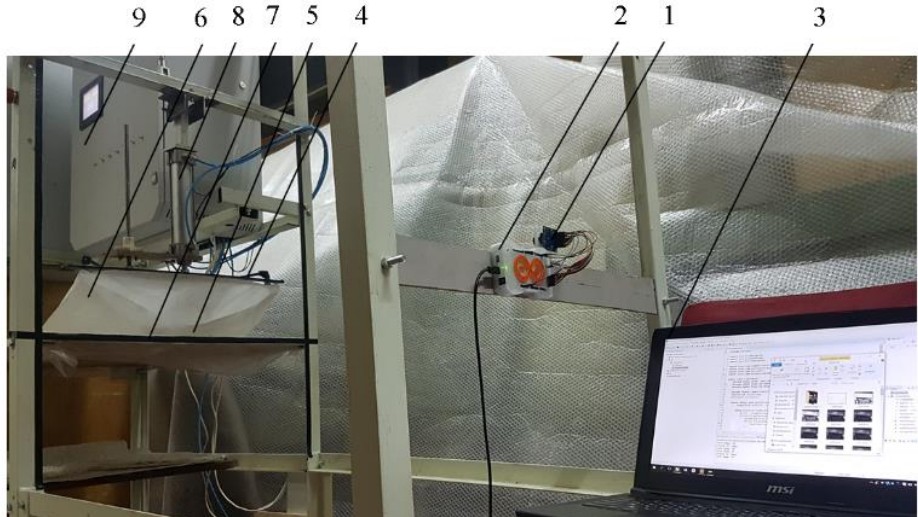

**Figure 14.** General view of the laboratory unit: 1, Raspberry Pi camera; 2, Raspberry Pi minicomputer; 3, laptop computer; 4, 5, vacuum gripping devices (VGDs); 6, flexible container; 7, table; 8, pneumatic cylinder; 9, control board.

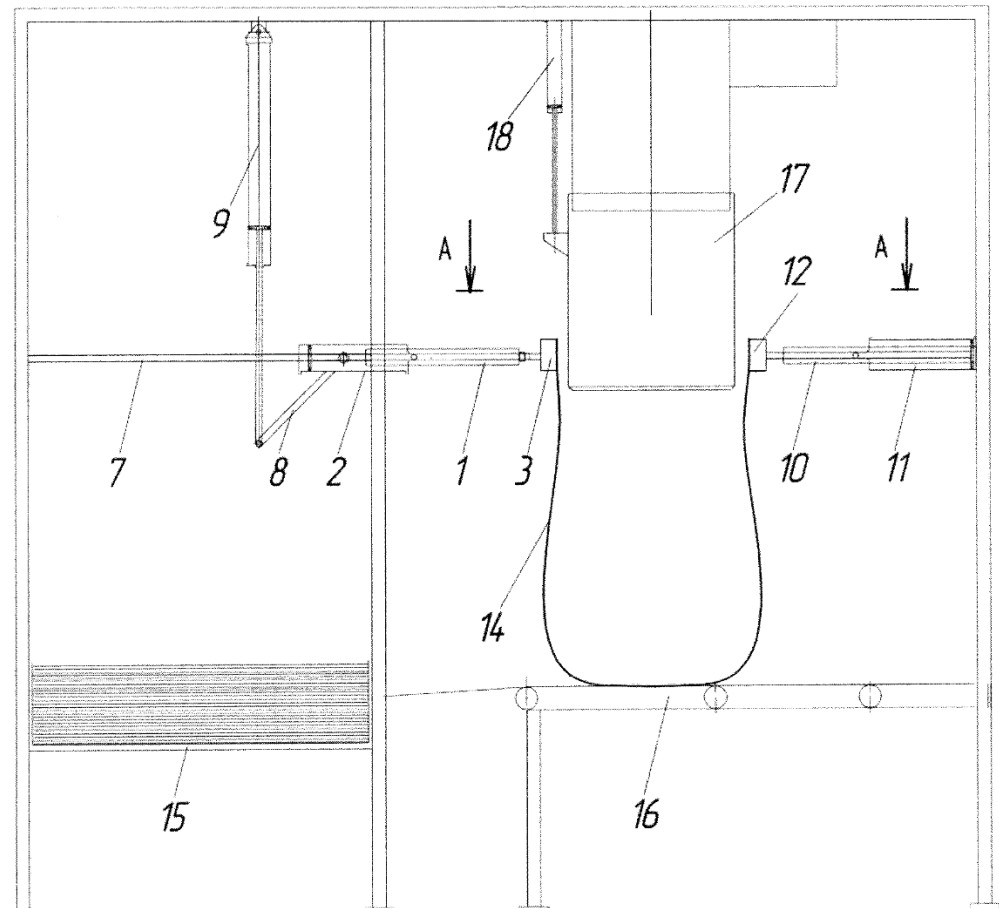

**Figure 15.** Scheme of the unit for automatic packaging of bulk materials.

A – A

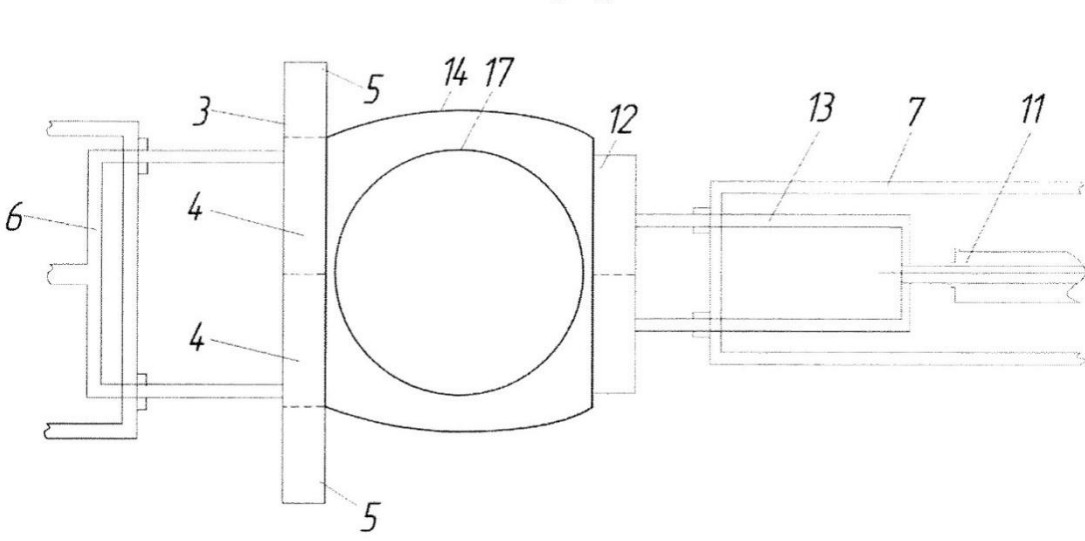

**Figure 16.** Scheme of the FC gripping and opening.

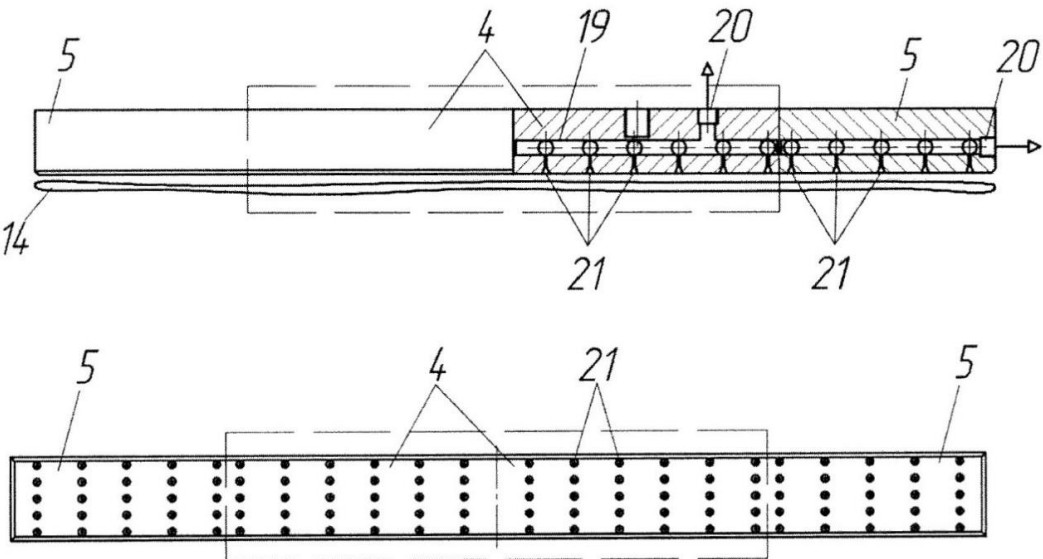

**Figure 17.** Scheme of the vacuum gripping device.

The opening mechanism (10) is mounted on the frame (7) opposite the gripping mechanism (1). It consists of a bilateral pneumatic cylinder (11) and a vacuum gripper (12), which is installed in a inverse manner to the central bars (4) of the vacuum gripper (3) of the gripping mechanism (1), attached to the U-shaped lever (13) and connected to the pneumatic cylinder (11). The stack of empty flexible containers (14), unstitched from their neck side, is located on the elevating platform (15) inside the frame (7). The conveyor belt (16) is located under the loading spout (17). The distance between the belt surface and the upper part of the vacuum grippers (3 and 12) of the gripping mechanism (1) and opening mechanism (10) is slightly less than the length of the filled flexible container (14). The loading spout (17) is attached to the bilateral pneumatic cylinder (18) and is capable of linear vertical relocation.

During the initial stage, the gripping mechanism (1) and the opening mechanism (10) are adjusted depending on the size of the flexible containers (14). The adjustment sets the required number of bars, the distance between the vacuum grippers (3 and 12), and the stroke length of the pneumatic cylinder (2 and 11) rods. It is carried out in such a way as to provide equal magnitudes of stroke and speeds of rod movement.

The adjustment of the pneumatic cylinder (9) rod stroke magnitude sets the end positions of the pneumatic cylinder (2) rod stroke. The end positions of the loading spout (17) are set by adjusting the pneumatic cylinder (18) rod stroke magnitude. If necessary and depending on the packaging cycle, the speed and lifting height of the platform (15) are adjusted as well. The elevating platform (15) fixes the position of the top flexible container (14) in the stack. Operation of the conveyor belt (16), the gripping mechanism (1), the opening mechanism (10), and the loading spout (17) is synchronized. The full packaging cycle is set and performed by the control system (not represented in the figures).

In its initial condition, the stack of flexible containers (14) is located horizontally on the elevating platform (15) so that all unstitched necks of the flexible containers are facing the conveyor belt (16) and the loading spout (17). The pneumatic cylinder (2) is located horizontally (the rod of the pneumatic cylinder (9) is protracted), while the vacuum gripper (3), particularly its central bars (4), and the vacuum gripper (12) are located in an inverse manner. The loading spout (17) is lifted.

When another cycle starts, the pneumatic cylinder (9) rod retracts, and the pneumatic cylinder (2) with the gripping mechanism (1) is turned 90 degrees clockwise. The vacuum gripper (3) is located above the necks of the stacked flexible containers (14). When the rod of the pneumatic cylinder (2) protracts, the vacuum gripper (3) presses itself to the neck of the container (14) which lies on top of the stack. A vacuum is created between the surface of the flexible container (14) and the vacuum chambers (21) of the bars (4 and 5) of the vacuum gripper (3). The vacuum gripper (3) moves to its initial position (the pneumatic cylinder (2) rod retracts), causing separation of the top flexible container (14) from the stack. The lower part of its neck does not sag significantly due to the full-width gripping of the neck area.

Then, the pneumatic cylinder (2) with the gripping mechanism (1) is moved 90 degrees counter-clockwise into a horizontal position by protracting the pneumatic cylinder (9) rod. The vacuum gripper (3) is located in an inverse manner to the vacuum gripper (12) of the opening mechanism (10). Meanwhile, the rod of the pneumatic cylinder (2) with the gripping mechanism (1) and the rod of the pneumatic cylinder (11) with the vacuum gripper (12) protract. A vacuum is created in the vacuum chambers of the gripper (12), similar to the vacuum chambers of the other gripper (3). The vacuum chambers of the end bars (5) of the vacuum gripper (3) become disconnected from the vacuum source. Pressure equal to ambient pressure starts to build up in the chambers, so they release the edges of the container (14) neck.

The neck of the flexible container (14) is opened by simultaneous retraction of the pneumatic cylinders' (2 and 11) rods. The neck is held evenly all over its surface by the gripping mechanism (1) with the help of the vacuum chambers of the central bars (4) of its vacuum gripper (3) and by the opening mechanism (10) with the help of the vacuum chambers of its vacuum gripper (12). The foundation of the flexible container (14) is located on the conveyor belt (16) directly under the loading spout (17), which sinks into the neck of the flexible container (14) and reaches the vacuum grippers (3 and 12) when the pneumatic cylinder (18) rod is pulled out. The preliminary weighed portion of bulk material starts to fill the container. When the filling is completed, the loading spout (17) is lifted to its initial position by retracting the pneumatic cylinder (18) rod. The vacuum grippers (3 and 12) move to each other and close the neck of the flexible container (14) as the rods of the pneumatic cylinders (2 and 11) are retracted. The grippers prepare the flexible container for capping (e.g. stitching). After the capping, the vacuum grippers (3 and 12) release the flexible container (14) by disconnection of the vacuum chambers (21) from the vacuum source. Located on the conveyor belt (16), the container (14) is moved to a shipment (storage) area. The rods of the pneumatic cylinders (2 and 11) are retracted. The elevating platform (15) lifts one position. The flexible container filling cycle is complete.

A block diagram of the experimental unit is presented in Figure 18.

Experiments checking the efficiency and accuracy of the developed diagnostic system were carried out on the basis of the automatic device for manipulating a flexible container during bulk material packaging [29,30]. The experimental unit contained a Raspberry Pi Camera Board v2.1 camera module with 3280 × 2464 px maximal resolution. Its images were processed by a Raspberry Pi 2 microcomputer

and were transferred to an MSI Gl62m 7rex 3 laptop using the special software. The camera lens was focused on the working area in which two vacuum gripping devices (4 and 5) and the flexible container (6) on the platform (7) were located [31].

The camera took pictures during the FC gripping process. Then, the pictures were transmitted to the Raspberry PI microcomputer, which further transmitted it via Ethernet to the special software for preliminary image processing and diagnostics of the FC gripping accuracy [32].

The VGD (4) consisted of several (at least two) vacuum chambers, the design of which provided for adjustment of the distance between end chambers and the possibility for vertical reciprocal motion using a Camozzi 60M2L040A0150 8 pneumatic cylinder. The other VGD (5) consisted of two vacuum chambers embedded into the platform (7) at a set distance. The unit was regulated by the control system installed in the control board (9). Motion of the VGD (4) was controlled by the 358-011-02 electropneumatic distributor (5/2, flow rate 700 Nl/min), while the vacuum degree in the vacuum chambers (4 and 5) was controlled by 623-15E-A6 electropneumatic distributors (valves) (2/2, flow rate 220 Nl/min) and SCV-20CK ejectors (vacuum up to −53.2 kPa, flow rate up to 180 Nl/min) which were connected to the compressed air feed system. The air was supplied from a GENESIS 11500 compressor. The operation sequence control was performed by a Mitsubishi Electric FX3U-16MT/DSS Electric [30,33] PLC.

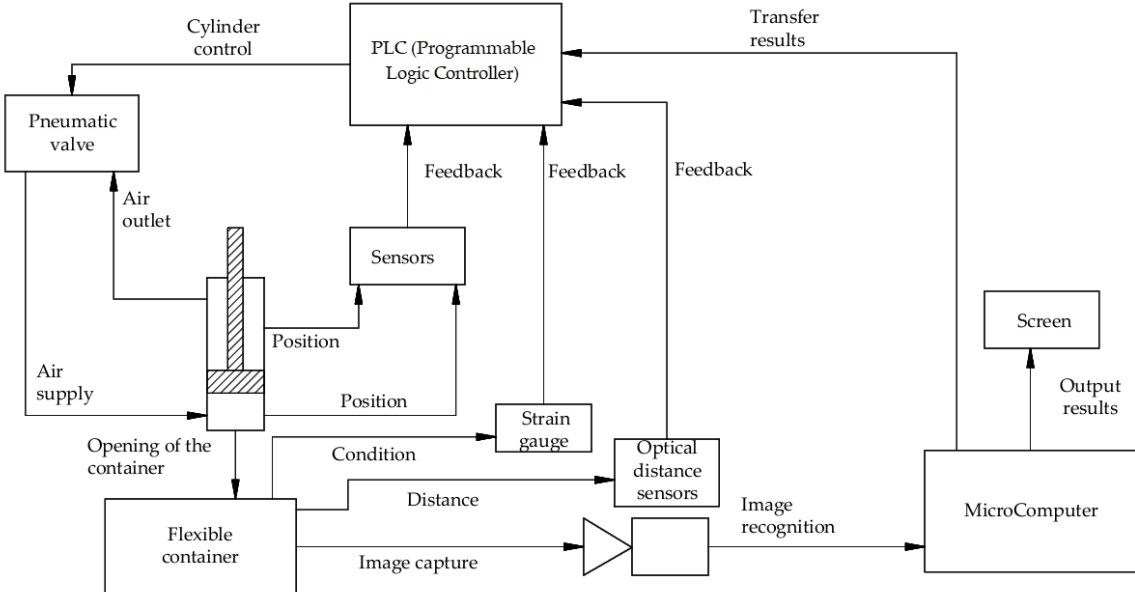

**Figure 18.** A block diagram of the experimental plant.

The experiment began when the START button was pressed. During the first stage, the VGD (4) sunk down until it touched the neck of the flexible container, which lay in the gripping position. After this, the vacuum chambers (VGD 4 and VGD 5) were depressurized. The vacuum gauge pointer indicated the current vacuum degree. When the VGD (4) returned to its starting position, the neck of the flexible container was slightly opened. A signal was sent to the microcontroller (2) and the camera took a picture which was transmitted to the laptop (3). Then, normal pressure was created in the vacuum chambers and the flexible container returned to its starting position. After that, the cycle was repeated. The experiment was considered finished when the STOP button was pressed or the set number of iterations was completed.

Standard flexible containers, woven from 4 mm wide polypropylene thread and with 50 kg capacity, were used for the experiment. The experiment was carried out under normal laboratory conditions, with the following initial parameters:

- flexible container dimensions: length, 1050 mm; width, 550 mm;
- stroke of the pneumatic cylinder piston with the upper VGD, 145 mm;
- diameter of the upper VGD vacuum chambers, 10 mm; distance between centers of the end side chambers, 470 mm.

## 4. Results and Discussion

As a result of the experiment on the preliminary opening of the FC, a series of photographs was obtained which was classified into several groups (Table 1). "Correct opening" means holding of the slightly opened FC neck form in the position which allows bulk material loading to freely fill the container. "Incorrect opening" does not fulfill the given condition and can be classified into several categories: the FC has gripped but opened incorrectly; the FC has not gripped; the FC has gripped but has not opened; the FC has gripped not by all vacuum chambers of the VGDs.

**Table 1.** Classification of photograph groups.

| Photograph | Classification | Photograph | Classification |
|---|---|---|---|
|  | Correct opening |  | Correct opening |
|  | Incorrect opening |  | Has not gripped |
|  | Has not opened |  | Has gripped not by all VGDs |

After that, the photographs were downloaded in the developed software (Figure 19) and were passed through the preliminary stage of processing; further, the ANN determined defects in the process of the FC gripping and opening which were then output in the report.

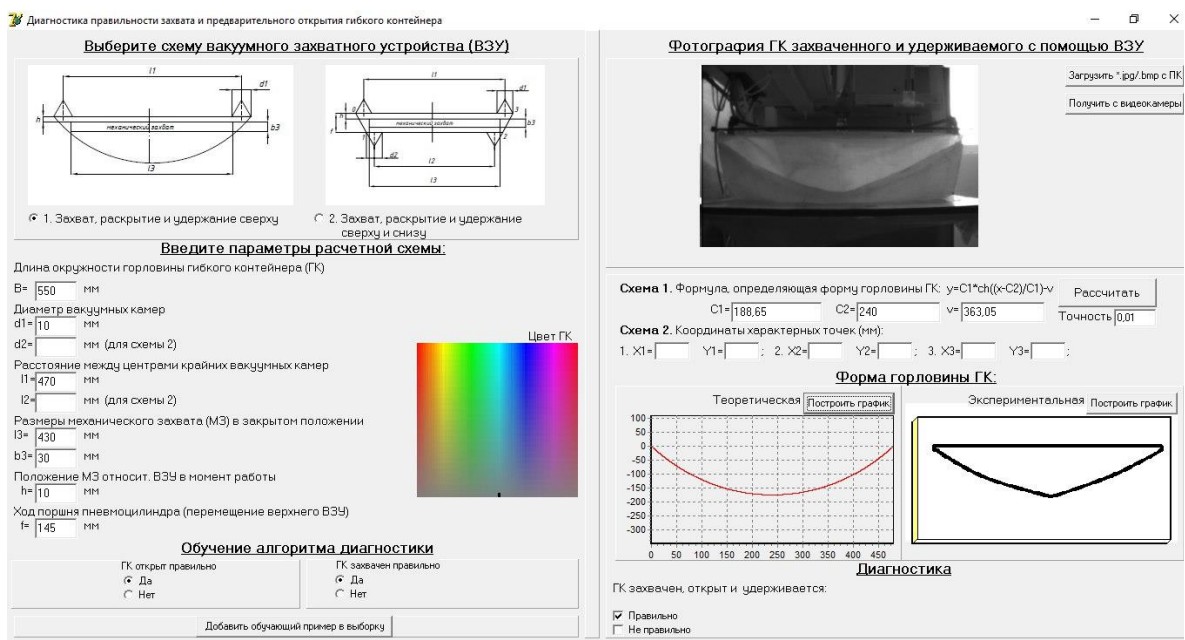

**Figure 19.** The working window of the developed software.

The determination error of the flexible container preliminary opening accuracy did not exceed 5% after training of the algorithm on a sample of not less than 100 photographs which included all marked classifications. Training of the algorithm allows for raising the precision of diagnostics.

The research results showed that when the flexible container is gripped and held only by the upper side, the probability of the flexible container incorrectly opening is high (more than 10% unsuccessful or incorrect grippings). It is expressed especially strongly when new FCs, which often have the effect of adhesion between their halves, are used. Some deviation between the experimental and theoretical forms of the opened neck was revealed when successful experiments were analyzed. It connected with a certain rigidness and did not have a significant influence on the successfulness of its following gripping from within with the help of, for example, a mechanical gripping device if its overall dimensions did not exceed the given quantity, which depends on the distance between the end points of holding.

## 5. Conclusions

A technique for the diagnostics and control of FC gripping accuracy in the process of the packaging of bulk materials was presented. This technique is based on the comparison of a picture obtained in the area of the FC gripping with a reference. Using the present technique, an experiment checking FC gripping and accuracy of its opening according to the theoretical (mathematical) model was carried out.

The functional model of the diagnostics process of the FC gripping and opening accuracy was developed in IDEF0 notation in terms of the represented algorithm. This model allows for considering the stages of diagnostics carried out in detail.

Technical development of the automated system of diagnostics of FC gripping and opening accuracy with the use of machine vision technology and a neural network algorithm processing the obtained pictures was described. The hardware component was realized on the basis of a single-board Raspberry Pi computer. The program suite was written in the Python programming language.

In the context of the further research in the present field, it is planned to secure the developed machine vision system with the possibility of remote control by means of a network-connected computer or human–machine interface (HMI) and also with the possibility of data exchange between the machine vision system and robots of any type or model which are used at the factories. Besides this, modification of the obtained system is intended for performance with any type or dimensions of packages.

## 6. Patents

1. Makarov, A.M.; Kristal', M.G.; Kovalev, A.A.; Sulejmanov, D.A. Device for automatic opening, holding and closing of flexible containers. RU 155000, 20 September 2015.

2. Makarov, A.M.; Lapikov, M.A.; Mushkin, O.V.; Serdobintsev, Y.P. Device for automatic gripping, opening and holding of flexible containers. RU 169411, 16 March 2017.

3. Makarov, A.M.; Mushkin, O.V.; Drobotov, A.V.; Serdobintsev, Y.P.; Kukhtik, M.P.; Lapikov M.A. Device for automatic opening, holding and closing of flexible containers. RU 186346, 16 January 2019.

**Author Contributions:** Project administration, A.M.M.; supervision, A.M.M.; writing—review and editing, O.V.M.; software, M.A.L.; writing—original draft, M.P.K.; formal analysis, Y.P.S.

**Funding:** This research received no external funding.

**Acknowledgments:** The authors would like to acknowledge administrative and technical support provided by the Volgograd State Technical University.

**Conflicts of Interest:** The authors declare no conflict of interest.

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
