# Peer review of "The Quality Control of the Automatic Manipulating Process of a Flexible Container When Bulk Materials are Packaged"

_machines, doi:10.3390/machines7040062_

Round 1

Reviewer 1 Report

In general, the paper is interesting and could be a good contribution to the topic. However, the manuscripts should be further revised improving the quality of presentation and the novelty of studies.

1) The introduction section should be improved. The recent trends in packaging technologies, diagnosis and the combination of different diagnostic systems should be reviewed, which will improve the novelty of the present paper. Some very recent research articles should be considered as well:

Metal Oxide Nanostructures in Food Applications: Quality Control and Packaging, Chemosensors 2018, 6(2), 16; https://doi.org/10.3390/chemosensors6020016

A review on innovations in polymeric nanocomposite packaging materials and electrical sensors for food and agriculture, Composite Interfaces 2019, https://doi.org/10.1080/09276440.2019.1600972

Recently emerging trends in polymer nanocomposites packaging materials, Polymer-Plastics Technology and Materials 2019, 58, 1054, https://doi.org/10.1080/03602559.2018.1542718

2) The system of automatic diagnostics of FC gripping accuracy should be described better.

3) The integration of the presented algorithm in packaging control systems should be described more in detail. Would be better to compare with other studies showing the novelty of the present work.

Reviewer 2 Report

Section 2. Authors should bring the numbering of formulas and decoding to them in accordance (on page 9 - formula number 3, on page 10 - formula number 4 and formula number 5). These numbers were on pages 4 line 119, 121, 122.

Authors have done good and artual for the community researsh
In the 2 section authors should check the formulas' numbering (the formulas are different, but the numbering are the same)

Section 3 are described efficiently and extensively. Requires a little spell check and English.

Section 4. I reccomended describe in detail Table 1, directions for further research or possible confines. The Discussion section exists, but it should be completed.

The author points out a large 23% of self-citations in the list of references; perhaps it should be reduced.

Round 2

Reviewer 1 Report

The quality of the paper has been improved. The manuscript can be published after the minor revision.

My previous comment "The system of automatic diagnostics of FC gripping accuracy should be described better" is related to section 2.4.

The authors reported detailed information on an eight-megapixel camera. So, may the photographs used by the software (eight- or ten-megapixel) have a big effect on the precision of diagnostics?

Has the connection mode of cameras such a big effect on the precision as well?
